# LEARNING DISCRETE STRUCTURED VARIATIONAL AUTO-ENCODER USING NATURAL EVOLUTION STRATEGIES

**Alon Berliner**
Technion, IIT
alon.berliner@gmail.com

**Guy Rotman**
Technion, IIT
rotmanguy@gmail.com

**Yossi Adi**
Meta AI Research
adiyoss@fb.com

**Roi Reichart**
Technion, IIT
roiri@technion.ac.il

**Tamir Hazan**
Technion, IIT
tamir.hazan@technion.ac.il

## ABSTRACT

Discrete variational auto-encoders (VAEs) are able to represent semantic latent spaces in generative learning. In many real-life settings, the discrete latent space consists of high-dimensional structures, and propagating gradients through the relevant structures often requires enumerating over an exponentially large latent space. Recently, various approaches were devised to propagate approximated gradients without enumerating over the space of possible structures. In this work, we use Natural Evolution Strategies (NES), a class of gradient-free black-box optimization algorithms, to learn discrete structured VAEs. The NES algorithms are computationally appealing as they estimate gradients with forward pass evaluations only, thus they do not require to propagate gradients through their discrete structures. We demonstrate empirically that optimizing discrete structured VAEs using NES is as effective as gradient-based approximations. Lastly, we prove NES converges for non-Lipschitz functions as appear in discrete structured VAEs.

## 1 INTRODUCTION

Discrete variational auto-encoders (VAEs) are able to represent structured latent spaces in generative learning. Consequently VAEs drive extensive research in machine learning applications, including language classification and generation [60, 17, 54, 9, 13], molecular synthesis [28, 15, 48], speech and visual understanding [36, 55, 3]. Compared to their continuous counterparts, they can improve interpretability by illustrating which terms contributed to the solution [48, 40], and they can facilitate the encoding of inductive biases in the learning process, such as images consisting of a small number of objects [12] or tasks requiring intermediate alignments [36, 42, 1, 2].

Learning VAEs with discrete $n$-dimensional latent variables is computationally challenging since the size of the support of the posterior distribution may be exponential in $n$. This is particularly common under the structured settings, when the latent variables represent complex structures such as trees or graphs. The Gumbel-max reparametrization trick trades enumeration with optimization using efficient dynamic programming algorithms and enables a computation of the model value. Unfortunately, the resulting mapping remains non-differentiable due to the presence of $\arg\max$ operations. In order to propagate gradients efficiently, Jang et al. [19], Maddison et al. [34] proposed the Gumbel-softmax reformulation that uses a smooth relaxation of the reparametrized objective, replacing the $\arg\max$ operation with a *softmax* operation. Following such an approach may bring back the need for enumerating over a large search space. This is due to the partition function of the softmax operator, which relies on a summation over all possible latent assignments, which may be exponential in $n$. To better deal with the computational complexity in the structured setting, sophisticated stochastic softmax tricks were devised to learn discrete structured VAEs [48] (e.g., perturb-and-parse for dependency parsing by [9], Gumbel-Sinkhorn for bi-partite matching [36]).

In this work we propose to use the Natural Evolution Strategy (NES) [57, 58] algorithm for learning discrete structured VAEs. The NES algorithm is a gradient-free black-box optimization method that does not need to propagate gradients through discrete structures. Instead, the NES algorithm estimates gradients by forward-pass evaluations only. We experimentally show that gradient-free methods are as effective as sophisticated gradient based methods, such as perturb-and-parse. NES is conceptually appealing when considering discrete structured VAEs since NES does not require to construct complex solutions to propagate gradients through the $\arg\max$ operation, as it only requires to evaluate the model. Moreover, the proposed approach is highly parallelizable, hence computationally appealing.

**Our contributions:** (1) We suggest using black-box, gradient-free based optimization methods, specifically NES, to optimize discrete structured VAEs. (2) We experimentally demonstrate that NES, which uses the models' output in a black-box manner, is as effective as gradient based approximations although being more general and simpler to use. (3) We rigorously describe the connection between NES and previous gradient based optimization methods (i.e, REINFORCE) as well as prove that the NES algorithm converges for non-Lipschitz functions.

## 2 BACKGROUND

**Discrete Structured Variational Auto-Encoders (VAEs)** learn a generative model $p_\theta(x)$ using a training set $S = \{x_1, \ldots, x_m\}$, derived from an unknown distribution $p(x)$ by minimizing its negative log-likelihood. VAEs rely on latent variable models of the form $p_\theta(x) = \sum_{z \in \mathcal{Z}} p_\theta(z)p_\theta(x|z)$, where $z$ is a realization of the latent variable and $\mathcal{Z}$ is the discrete set of its possible assignments. We focus on discrete structures such as spanning trees in a graph $G = (V, E)$, i.e., $z = (z_1, \ldots, z_{|E|})$ represents a spanning tree $T$ for which $z_e = 1$ if the edge $e \in T$ belongs to the spanning tree $T$ and zero otherwise. In this case, $\mathcal{Z}$ is the spanning trees space which is typically exponential in the size of the input, as there are $|V|^{|V|-2}$ spanning trees for a complete graph with $|V|$ vertices.

VAEs rely on an auxiliary distribution $q_\phi(z|x)$ that is used to upper bound the negative log-likelihood of the observed data points: $\sum_{x \in S} -\log p_\theta(x) \leq \sum_{x \in S} L(\theta, \phi, x)$, where: $L(\theta, \phi, x) = -\mathbb{E}_{z \sim q_\phi(\cdot|x)}[\log p_\theta(x|z)] + KL(q_\phi(\cdot|x)||p_\theta(\cdot))$. This formulation is known as the negative Evidence Lower Bound (ELBO) [20], where the KL-divergence measures the similarity of two distributions $q_\phi$ and $p_\theta$, and is defined as: $KL(q_\phi(\cdot|x)||p_\theta(\cdot)) = \mathbb{E}_{z \sim q_\phi(\cdot|x)}[\log(q_\phi(z|x)/p_\theta(z))]$.

Parameter estimation is generally carried out by performing gradient descent on $\sum_{x \in S} L(\theta, \phi, x)$. In the discrete VAE setting, the first term admits an analytical closed form gradient:

$$\frac{\partial \mathbb{E}_{z \sim q_\phi(\cdot|x)}[\log p_\theta(x|z)]}{\partial \phi} = \mathbb{E}_{z \sim q_\phi(\cdot|x)}\left[\log p_\theta(x|z)\frac{\partial \log q_\phi(z|x)}{\partial \phi}\right]. \tag{1}$$

An exact computation of the expectation requires enumeration over all possible latent assignments since $\mathbb{E}_{z \sim q_\phi(\cdot|x)}[\log p_\theta(x|z)] = \sum_{z \in \mathcal{Z}} q_\phi(z|x)\log p_\theta(x|z)$. Unfortunately, in the structured setting, the number of possible latent assignments is exponential. Instead, one can rely on the score function estimator (REINFORCE) to generate an unbiased estimate of the gradient by sampling from the distribution over the latent space. In many cases of interest, such as sampling spanning trees, the sampling algorithm is computationally unfavorable and suffers from high variance, leading to slow training and poor performance [46].

The Gumbel-Max reparametrization trick can trade summation with optimization. This approach is computationally appealing when considering spanning trees, since finding a maximal spanning tree is more efficient than sampling a spanning tree. Consider i.i.d. zero-location Gumbel random variables $\gamma \sim \mathcal{G}(0)$, e.g., in the case of spanning trees $\gamma = (\gamma_1, \ldots, \gamma_{|E|})$ consists of an independent random variable for each edge in the graph. Let $q_\phi(z|x) = e^{z^\top h_\phi(x)}$ where $h_\phi(x)$ is a parametric encoder that learns edge scores and $p_\theta(x|z) = e^{f_\theta(x,z)}$, where $f_\theta(x, z)$ denotes the log-probability learned by a parametric decoder. Then, the summation $\sum_{z \in \mathcal{Z}} -q_\phi(z|x)\log p_\theta(x|z)$ can be approximated by the expectation

$$\mathbb{E}_{\gamma \sim \mathcal{G}(h_\phi(x))}\left[-f_\theta(x, \arg\max_{z \in \mathcal{Z}}\{z^\top\gamma\})\right]. \tag{2}$$

We provide the derivation in Appendix C. This formulation is a key to our proposed approach, as in some cases, estimating the above equation is easier. Computing the $\arg\max$ can be accomplished

efficiently even when the latent space is exponentially large. It is performed by utilizing sophisticated MAP solvers. For instance, finding the maximum spanning tree can be achieved in polynomial run time using Kruskal's algorithm [27]. Sampling by perturbing the input and feeding it to a MAP solver is called perturb-and-map [47]. The gradient of Eq. 2 can be estimated using REINFORCE without needing to relax the $\arg\max$ operator. By reparametrizing the Gumbel distribution, we get:

$$\frac{\partial \mathbb{E}_{\gamma \sim \mathcal{G}(h_\phi(x))}\left[f_\theta(x, \arg\max_{z \in \mathcal{Z}}\{z^\top \gamma\})\right]}{\partial \phi} = \mathbb{E}_{\gamma \sim \mathcal{G}(h_\phi(x))}\left[f_\theta(x, \arg\max_{z \in \mathcal{Z}}\{z^\top \gamma\})\frac{\partial \log \mathcal{G}(h_\phi(x))(\gamma)}{\partial \phi}\right]. \quad (3)$$

Where $\mathcal{G}(h_\phi(x))$ denotes the probability density function (PDF) of the Gumbel distribution with a location parameter of $h_\phi(x)$. The exponential summation in Eq. 1 is replaced with optimization, and now samples can be derived efficiently by applying the perturb-and-map technique. On the other hand, the disadvantages of REINFORCE remain as they were.

The Gumbel-Softmax trick is a popular approach to reparameterize and optimize Eq. 2. Since $\mathbb{E}_{\gamma \sim \mathcal{G}(h_\phi(x))}\left[f_\theta(x, \arg\max_{z \in \mathcal{Z}}\{z^\top \gamma\})\right] = \mathbb{E}_{\gamma \sim \mathcal{G}(0)}\left[f_\theta(x, \arg\max_{z \in \mathcal{Z}}\{z^\top(h_\phi(x) + \gamma)\})\right]$, one can apply the Softmax to replace the non-differential $\arg\max$ function. Hence, the Gumbel-Softmax trick replaces the function $\arg\max_{z \in \mathcal{Z}}\{z^\top(h_\phi(x) + \gamma)\})$ with the differential softmax function $\frac{e^{z^\top(h_\phi(x)+\gamma)}}{\sum_{\hat{z} \in \mathcal{Z}} e^{\hat{z}^\top(h_\phi(x)+\gamma)}}$, cf. Jang et al. [19], Maddison et al. [34]. Under the structured setting, sophisticated extensions avoid the exponential summation in the partition function of the softmax operator [9, 36, 48]. For instance, in dependency parsing, Corro & Titov [9] construct the differentiable perturb-and-parse (DPP) method that exploits a differentiable surrogate of the Eisner algorithm [11] for finding the highest-scoring dependency parsing by replacing each local $\arg\max$ operation with a softmax operation. Alternatively, Paulus et al. [48] utilize the Matrix-Tree theorem [24] for propagating approximated gradients through the space of undirected spanning trees.

**Natural Evolution Strategies (NES)** is a class of gradient-free optimization algorithms. NES optimizes its objective function, by evaluating it at certain points in the parameter space. Consider a function $k(\mu)$, may it be non-differentiable nor continuous, instead of optimizing $k(\mu)$ using a gradient method, NES optimizes a smooth version using the expected parameters of the function:

$$g(\mu) = \mathbb{E}_{w \sim \mathcal{N}(\mu, \sigma^2 I)}[k(w)] = \int_{\mathbb{R}^d} \frac{1}{(2\pi\sigma^2)^{\frac{d}{2}}} e^{-\frac{\|w-\mu\|^2}{2\sigma^2}} k(w) dw. \quad (4)$$

Here $\mathcal{N}(\mu, \sigma^2 I)$ is a Gaussian distribution with mean $\mu$ and covariance $\sigma^2 I$. The expectation with respect to the Gaussian ensures the function $g(\mu)$ is differentiable, since $e^{-\frac{\|w-\mu\|^2}{2\sigma^2}}$ is differentiable of any order, although $k(\mu)$ may not be differentiable. Following the chain-rule, the score function estimator for the Gaussian distribution determines the gradient:

$$\frac{\partial g(\mu)}{\partial \mu} = \int_{\mathbb{R}^d} \frac{1}{(2\pi\sigma^2)^{\frac{d}{2}}} e^{-\frac{\|w-\mu\|^2}{2\sigma^2}} \left(\frac{w-\mu}{\sigma^2}\right) k(w) dw. \quad (5)$$

That is, NES is an instance of REINFORCE, which optimizes a smoothed version of $k(\mu)$ by sampling from a distribution over parameter space rather than latent space. The reparameterization trick allows to further simplify the gradient estimator, with respect to a standard normal distribution:

$$\frac{\partial g(\mu)}{\partial \mu} = \int_{\mathbb{R}^d} \frac{1}{(2\pi)^{\frac{d}{2}}} e^{-\frac{\|w\|^2}{2}} \left(\frac{w}{\sigma}\right) k(\mu + \sigma w) dw = \mathbb{E}_{w \sim \mathcal{N}(0, I)}\left[\frac{w}{\sigma} k(\mu + \sigma w)\right]. \quad (6)$$

The gradient can be estimated by sampling $w$ repeatedly from a standard Gaussian: $\frac{\partial g(\mu)}{\partial \mu} \approx \frac{1}{N}\sum_{i=1}^{N} \frac{w_i}{\sigma} k(\mu + \sigma w_i)$. The obtained estimator is biased when $\sigma > 0$. However, the bias approaches 0 as $\sigma \to 0$. In practice, $\sigma$ is assigned a small value, treated as a hyper-parameter.

This algorithm is computationally appealing as it only uses the evaluations of $k(\cdot)$ to compute the gradient of $g(\cdot)$. Moreover, NES is highly parallelizable, i.e., one can compute the gradient $\nabla g(\mu)$ at the time of evaluating a single $k(\mu + \sigma w_i)$, in parallel for all $i = 1, ..., N$, and then average these parallel computations [52]. Figure 1 depicts the parallel forward passes and the update rule according to NES.

Theoretical guarantees regarding the convergence of gradient-free methods such as NES, were established for Lipschitz functions by Nesterov & Spokoiny [41]. In Section 3, we extend the current zero-order optimization theory by presenting a convergence bound for NES over non-Lipschitz and bounded functions.

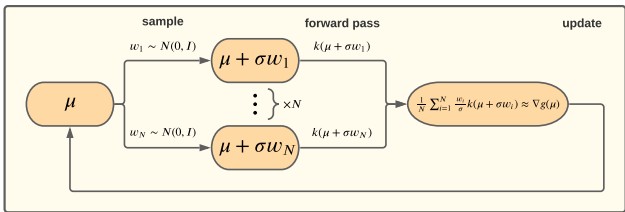

Figure 1: NES perturbs the parameters of a black-box model $k(\mu)$ for $N$ times via $w, \sigma$ and performs parallel forward passes. Then, the model parameters are updated given the gradient estimation.

## 3 STRUCTURED VAE OPTIMIZATION USING NES

In this work we suggest using gradient-free based method for learning discrete structured VAEs. Specifically, we propose using NES to optimize discrete structured VAEs without the need to propagate gradients through their latent discrete structures. For readability, we consider a concatenation of both $\theta$ and $\phi$ as $\mu = [\theta; \phi]$ where ; is the concatenation operation. For notational convenience, we refer $\theta$ as $\mu_1$ and $\phi$ as $\mu_2$.

Combining the discrete VAE objective function in Eq. 2, which is a non-continuous function due to the $\arg \max$ operation and hence, non-Lipschitz, together with the NES objective in Eq. 4 (i.e., setting $k(\mu) = L(\mu, x)$) we get the following smooth approximation by setting $g(\mu)$ to be:

$$\mathbb{E}_{w \sim \mathcal{N}(\mu, \sigma^2 I)} \mathbb{E}_{\gamma \sim \mathcal{G}(h_{w_2}(x))} \Big[ - f_{w_1}(x, \arg \max_{z \in \mathcal{Z}} \{z^\top \gamma\}) \Big], \tag{7}$$

where $w = [w_1; w_2]$ is the concatenation of vectors $w_1, w_2$ that denotes the decoder and encoder parameters respectively. Our goal is to minimize Eq. 7. Following the NES setup, we use Eq. 6 with a simplified notation to better emphasize the smoothing of $\mu$. Overall its gradient takes the form of:

$$\mathbb{E}_{w \sim \mathcal{N}(0, I)} \mathbb{E}_{\gamma \sim \mathcal{G}(h_{\mu_2 + \sigma w_2}(x))} \Big[ - \frac{w}{\sigma} f_{\mu_1 + \sigma w_1}(x, \arg \max_{z \in \mathcal{Z}} \{z^\top \gamma\}) \Big]. \tag{8}$$

We provide the pseudo code for using the NES algorithm to optimize discrete VAEs, together with their gradient update rule on Algorithm 1. We estimate the NES gradient in Eq. 8 by sampling $w \sim N(0, I)$ and $\gamma \sim \mathcal{G}(h_{\mu_2 + \sigma w_2}(x))$, which in turn induces a sampling of discrete structures $\arg \max_{z \in \mathcal{Z}} \{z^\top \gamma\}$. This sampling procedure differs from that of the REINFORCE instances described in Eq. 1 and Eq. 3, as the samples of NES are tied to the sensitivity of the scoring function $h_{\mu_2 + \sigma w_2}(x)$, i.e, to a random perturbation of its parameters $\mu_2$ by $\sigma w_2$. In contrast, the samples of REINFORCE are proportional to the scoring function $h_{\mu_2}(x)$ itself. In our experimental validation, we empirically demonstrate that NES has a lower variance.

**Theoretical Guarantees.** Next, we prove NES converges for a non-Lipschitz function $k(\cdot)$, which appears in reparameterized discrete VAEs in Eq. 2. In particular, we show that the norm of the parameters' gradient can be arbitrarily small as training progresses. More formally, when the NES algorithm performs $T$ update rules to the parameters $\mu$, it generates the sequence $\mu^{(1)}, \mu^{(2)}, ..., \mu^{(T)}$, and for a sufficiently large $T$, there exists $t \in \{1, ..., T\}$ for which $\|\nabla g(\mu^{(t)})\|$ is arbitrarily small.

For mathematical simplicity, we prove our convergence theorem on the expected gradient, as described in Eq. 8: $\mu^{(t+1)} \leftarrow \mu^{(t)} - \eta \nabla g(\mu^{(t)})$. To easily address $\|\nabla g(\mu^{(t)})\|$ one usually considers the difference $\mu^{(t+1)} - \mu^{(t)}$ using the remainder of the Taylor series: $g(\mu^{(t+1)}) - g(\mu^{(t)}) = \nabla g(\mu^{(t)})^\top (\mu^{(t+1)} - \mu^{(t)}) + \frac{1}{2}(\mu^{(t+1)} - \mu^{(t)})^\top \nabla^2 g(\hat{\mu})(\mu^{(t+1)} - \mu^{(t)})$, where $\hat{\mu} \in [\mu^{(t)}, \mu^{(t+1)}]$. Here we denote by $\nabla^2 g(\hat{\mu})$ the Hessian of $g(\hat{\mu})$. Applying the gradient update rule, we obtain the following equation:

$$g(\mu^{(t+1)}) - g(\mu^{(t)}) = -\eta \|\nabla g(\mu^{(t)})\|^2 + \frac{\eta^2}{2} \nabla g(\mu^{(t)})^\top \nabla^2 g(\hat{\mu}) \nabla g(\mu^{(t)}). \tag{9}$$

A bound for $\nabla g(\mu^{(t)})^\top \nabla^2 g(\hat{\mu}) \nabla g(\mu^{(t)})$ is a key to our convergence theorem. Our bound relies on the fact that the discrete VAE objective, which is given in Eq. 2, is a non-negative function that is continuous almost everywhere.

---

**Algorithm 1** Natural Evolution Strategies for discrete VAEs

---
**Input:** Initial parameters $\mu$.
**repeat**
    **for all** $i = 1$ **to** $N$ **do**
        Sample $\tilde{w}_i \sim \mathcal{N}(0, I_{|\mu|})$
        Evaluate $u_i = \mathbb{E}_{\gamma \sim \mathcal{G}(h_{\mu_2 + \sigma \tilde{w}_{2,i}}(x))}\Big[ -f_{\mu_1 + \sigma \tilde{w}_{1,i}}(x, \arg\max_{z \in \mathcal{Z}}\{z^\top \gamma\}) \Big]$
    **end for**
    Update $\mu \leftarrow \mu - \eta \cdot \frac{1}{N} \sum_{i=1}^{N} \frac{\tilde{w}_i}{\sigma} u_i$
**until** a stopping condition is met

---

**Lemma 1.** *Let $k : \mathbb{R}^d \to [0, M]$ be a non-negative function that is continuous almost everywhere and let $g(\mu) = \mathbb{E}_{w \sim \mathcal{N}(\mu, \sigma^2 I)}[k(w)]$. Then for any $\mu_1, \mu_2$ there holds:*

$$\nabla g(\mu_1)^\top \Big( \nabla^2 g(\mu_2) \Big) \nabla g(\mu_1) \leq \frac{dM^3}{\sigma^4}. \tag{10}$$

Proof can be found in Appendix A.1. The above lemma together with Eq. 9 imply the following bound: $g(\mu^{(t+1)}) - g(\mu^{(t)}) \leq -\eta \|\nabla g(\mu^{(t)})\|^2 + \eta^2 \frac{dM^3}{2\sigma^4}$. By summing over all algorithm steps $t = 1, ..., T$ we show that the average norm of the gradient can be arbitrarily small for a sufficiently large $T$.

**Theorem 1.** *Under the conditions of Lemma 1 there holds:*

$$\frac{1}{T} \sum_{t=1}^{T} \|\nabla g(\mu^{(t)})\|^2 \leq \frac{M}{\eta T} + \frac{\eta dM^3}{2\sigma^4}. \tag{11}$$

*Moreover, when setting $\eta = \sqrt{\frac{2\sigma^4}{T d M^2}}$ then: $\frac{1}{T} \sum_{t=1}^{T} \|\nabla g(\mu^{(t)})\|^2 \leq \sqrt{\frac{2dM^4}{T\sigma^4}}$.*

*Therefore, there exists $t$ for which $\|\nabla g(\mu^{(t)})\|^2 \leq \sqrt{\frac{2dM^4}{T\sigma^4}}$.*

Proof is given in Appendix A.2. Intuitively, the above theorem proves that the NES algorithm converges on discrete VAE to a stationary point, even when the original function $k(\cdot)$ is non-continuous and hence non-Lipschitz. This is in contrast to the contemporary trend that relies on Lipschitz functions [41]. Instead, we rely on the non-negativity of the discrete VAE objective. We note that a stationary point of $g(\mu) = \mathbb{E}_{w \sim \mathcal{N}(\mu, \sigma^2 I)}[k(w)]$ is not necessarily a stationary point of $k(\mu)$. Nevertheless, since $\lim_{\sigma \to 0} g(\mu) = k(\mu)$ almost everywhere, except perhaps for non-continuous points, a low value of $g(\mu)$ is correlated with a low value of $k(\mu)$.

## 4 EXPERIMENTS

We start by experimentally validating our approach by learning discrete structured VAEs for latent structure recovery in Section 4.1 and dependency parsing in Section 4.2. Next, in Section 4.3 we analyze how NES scales with the latent space dimension and neural network size. We additionally provide an analysis for non-Lipschitz functions in Appendix B, together with analyzing the results concerning the theoretical guarantees as presented in Section 3.

In our experiments, we use a variance reduction technique called *mirrored sampling* [14, 4]: that is, on each NES iteration, we use a single Gaussian noise vector $w$ to create two parameter sets, one by adding and the other by subtracting the noise vector. Thus, on each iteration, we sample $\frac{N}{2}$ Gaussian noise vectors where $N$ is the number of the VAE parameter sets utilized for estimating the NES update direction. Additionally, to make NES more robust and scale-invariant, we transform the outputs of the perturbed forward passes into standard scores by subtracting their mean and dividing them by the standard deviation. All reported values are measured on a test set, and the models were selected using early stopping on the validation set. All the following experiments were conducted using an internal cluster with 4 Tesla-K80 NVIDIA GPUs.

### 4.1 LATENT STRUCTURE RECOVERY

We begin by demonstrating the capability of NES to learn the internal structure of an interacting system based on graphs in an unsupervised fashion. The interplay of group components, e.g., basketball players on the court or a flock of birds during migration, can often be explained using a

Table 1: The mean and standard deviation of the ELBO and structure recovery metrics. NES outperforms the five different baselines in terms of structure recovery and presents a negligible standard deviation, which reflects its robustness.

| Method | *Spanning Tree* | | $n-1$ *Individual Edges* | |
|---|---|---|---|---|
| | ELBO ↑ | Edge F1-Score ↑ | ELBO ↑ | Edge F1-Score ↑ |
| REINFORCE (*Batch*) | -2260 ± 0 | 41 ± 1 | -2180 ± 0 | 39 ± 1 |
| REINFORCE (*EMA*) | -2250 ± 20 | 40 ± 7 | -2170 ± 10 | 42 ± 1 |
| REINFORCE (*Multi-sample*) | -2230 ± 20 | 42 ± 1 | -2150 ± 10 | 40 ± 0 |
| NVIL | -1570 ± 300 | 83 ± 20 | -2110 ± 10 | 42 ± 2 |
| SST | **-1080 ± 110** | 91 ± 3 | **-2100 ± 20** | 41 ± 1 |
| NES (*Ours*) | -1117 ± 45 | **92 ± 0.2** | -2150 ± 10 | **44 ± 0.6** |

simple structure. However, frequently, we only have access to individual trajectories without knowledge of the underlying interactions. The Neural Relational Inference (NRI) model [23], which we base on, is designed to infer these interactions purely from observational data. NRI takes the form of a VAE, where the encoder produces a distribution over the space of interaction structures given the component trajectories, and the reconstruction is based on graph neural networks.

In our experiments, we utilize the dataset developed by Paulus et al. [48], where the target structure is a spanning tree over 10 vertices (each vertex represents an individual component). The model attempts to learn the true tree structure that defines the interplay among the group by only observing the locations of the 10 components during several timesteps. We focus on two cases, as suggested by Paulus et al. [48]. In the first case, we use our prior knowledge regarding the true structures and define the latent space as the space of spanning trees over a 10-nodes undirected graph, which consists of $10^8$ possible spanning trees. In the second more challenging case, we remove the tree constraint and consider all possible $n-1$ unique edge combinations as the latent space, where $n$ denotes the number of vertices. The ability to recover the underlying structure is measured as the edge F1-score against the target spanning tree.

In each of the two cases, we compare NES with the corresponding Stochastic Softmax Trick (SST) [48]. SSTs are the generalization of the Gumbel-Softmax Trick (GSM) for combinatorial discrete distributions. That is, in contrast to GSM, SSTs are designed to optimize over exponentially large discrete spaces. We run our experiments with the same set of parameters as in Paulus et al. [48], except that during decoding we use teacher-forcing every 3 steps instead of 9 steps. We fix NES parameters to be $\sigma = 0.01$ and $N = 600$. We additionally compared our method against four instances of REINFORCE. Each utilizes a different variance reduction technique. The first is NVIL [39], which uses two control variates. The remaining three reduce variance by subtracting the following control variate from the learning signal: *EMA* uses the exponential moving average of the ELBO, *Batch* uses the mean ELBO of the current mini-batch, and lastly, *Multi-sample*, which is especially well suited for structured prediction [25, 26], uses the mean ELBO over $r$ multiple samples per data point. Paulus et al. [48] tuned $r$ on the set of $\{2, 4, 8\}$. Results are listed in Table 1.

In terms of structure recovery, the gradient-free NES outperforms all REINFORCE instances. Moreover, when considering SST, NES achieves superior edge F1-score with slightly worse ELBO values. This is surprising since SST is a gradient-based method that generalizes the effective GSM estimator to combinatorial spaces. Also, unlike SST, which requires a carefully tailored solution for each structure, NES is simple and generic. For a fair comparison with NES, we also ran the *Multi-sample* method with 400 Monte Carlo samples per data point and a mini-batch size of 4 (as using larger $r$ or larger mini-batch size has exceeded the GPU memory limit). However, the results were inferior to those achieved by tuning $r$ on $\{2, 4, 8\}$. This is in line with the results obtained by Kool et al. [25], where larger values of $r$ led to inferior results. Thus, it can be concluded that even with an equal computational cost, the REINFORCE instances are inferior to NES.

## 4.2 DEPENDENCY PARSING

Next, we evaluate the capability of NES in learning latent projective and non-projective dependency parse trees as part of an unsupervised domain adaptation task. Unlike Section 4.1, where we focused on structures over undirected edges, here we focus on dependency trees which are rooted directed spanning trees. Our model is based on a VAE architecture similar to that of differentiable perturb-and-parse (DPP) [9]. The encoder is comprised of a graph-based parser [22] that decomposes the

Table 2: Unlabeled attachment scores for unsupervised domain adaptation. The column name represents the setup where the treebank is set to be the target domain. For each of the two tasks, NES achieves the best UAS performance on 5 out of the 6 target domains.

| Method | GL_CTG | GL_TREEGAL | ID_CSUI | ID_GSD | RU_GSD | RU_TAIGA | Projective |
|---|---|---|---|---|---|---|---|
| DPP | 68.72 | 71.39 | 68.23 | **71.71** | 71.46 | 69.94 | ✓ |
| NES (*Ours*) | **68.92** | **71.64** | **68.28** | 71.41 | **71.82** | **70.52** | ✓ |
| SparseMAP | 68.57 | 70.61 | 68.23 | **70.56** | 70.99 | **69.96** | ✗ |
| NES (*Ours*) | **68.67** | **70.98** | **68.28** | 70.47 | **71.01** | 69.96 | ✗ |

score of a tree to the sum of the scores of its arcs and produces a distribution over the space of dependency trees. Sampling from the latent space is performed using the "perturb-and-map" technique, where each arc score is perturbed independently with a noise derived from a Gumbel distribution. Then, the perturbed arc scores are fed into a MAP solver, which outputs the highest-scoring tree (the resulting sample). For projective dependency parsing, we utilize the Eisner algorithm [11] as the MAP solver. Similarly, for non-projective dependency parsing, we use the Chu-Liu-Edmonds (CLE) algorithm [6, 10]. The decoder is modeled as a language model, that given a latent dependency tree, attempts to reconstruct the input sentence.

We compare NES with two strong baselines. For projective dependency parsing, we consider the DPP model. DPP optimizes the VAE by utilizing a differentiable surrogate of the Eisner algorithm. In that manner, it tackles both the differentiability and exponential enumeration issues. In fact, DPP can be seen as a Stochastic Softmax Trick (SST). For non-projective dependency parsing, we consider SparseMAP [43] as a baseline by replacing the CLE algorithm with a SparseMAP layer. SparseMAP uses the active set method that performs sequential calls to a MAP solver and returns a sparse linear combination of several high-scoring structures. This procedure is differentiable almost everywhere but computationally inefficient due to its sequential nature. Unlike these methods, NES does not require any modification to the architecture.

We perform extensive experiments on the task of unsupervised domain adaptation for dependency parsing. We consider the Universal Dependencies (UD) dataset [35, 44, 45]. UD is a multilingual corpus annotated with dependency trees in more than 180 treebanks of over 100 languages. We follow the setup of Rotman & Reichart [50] and choose 3 distinct languages, considering 2 distinct treebanks from different domains for each: Galician (GL_CTG: science and legal, GL_TREEGAL: news), Indonesian (ID_CSUI: news, ID_GSD: general) and Russian (RU_GSD: general, RU_TAIGA: social media, poetry, and fiction). We conduct 6 domain adaptation experiments, where we alternate between the source and target domain in each language. We consider the training set of our source domain as our labeled dataset and the training set of the target domain as the unlabeled dataset.

At first, we train the VAE components separately on the labeled set (source domain) for 30 epochs. Then, we optimize the pretrained VAE on the unlabeled set (target domain) for 10 additional epochs using the NES algorithm. For a fair comparison, we perform the same training procedure for the above-mentioned baselines. We set the hyper-parameters to those of the original implementation of Kiperwasser & Goldberg [22] and feed the models with the multilingual FastText word embeddings [16]. We perform a grid-search for each of the methods separately over learning rates in $[5 \cdot 10^{-4}, 1 \cdot 10^{-5}]$ and set the mini-batch size to 128. We fix NES parameters to be $\sigma = 0.1$ and $N = 400$. Adam optimizer [21] is used to optimize all methods. The models we selected were those who obtained the best unlabeled attachment score (UAS) on the source domain validation set.

Table 2 summarizes the results on the UD treebanks in terms of unlabeled attachment score (UAS). The scores under each treebank name reflect performances on the setup where the treebank is set to be the target domain. Results suggest that NES reaches comparable performance (with a minor improvement) to SparseMAP and DPP while being simpler and more flexible to use. Note that unlike SparseMAP and DPP, which use sequentially complex methods to either infer the highest-scoring tree structure or to propagate gradients through a bottleneck dynamic programming algorithm, NES can optimize the model in parallel without the necessity of gradient computations.

## 4.3 SCALABILITY ANALYSIS

**Latent space size.** In the following set of experiments, we further investigate the properties of NES and the several methods it was compared to in Section 4.1 and 4.2. Specifically, we examine how

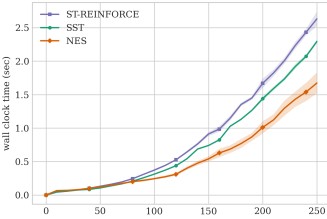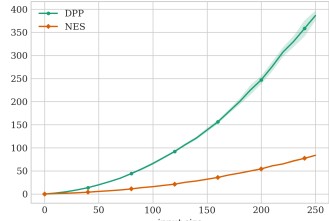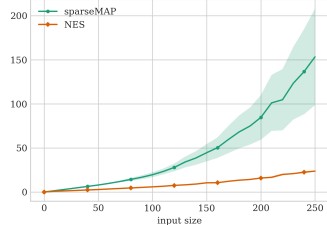

Figure 2: Wall-clock time as a function of model input size. Experiments are conducted on the NRI model (**Left**), projective parsing model (**Center**), and non-projective parsing model (**Right**). We observe that NES scales well with the latent space size in contrast to most of its competitors.

the latent space size affects the method's run-time by measuring the methods wall-clock time of a forward and backward pass as a function of the input dimension (denoted by $n$). Note that the latent space size grows exponentially with the model input dimension, e.g., for a sentence of length $n$, the latent space of the VAE architecture presented in Section 4.2 is the space of all possible dependency trees over an $n$-nodes directed graph.

For each $n \in \{10, 20, \dots, 250\}$ we create a random dataset. Specifically, for the NRI model (Section 4.1), the input is $n$ trajectories of 10 timesteps derived from a standard Gaussian distribution. We compare the run-time of NES with that of SST and a REINFORCE instance that relies on Eq. 1, where sampling is performed using a Markov chain Monte Carlo (MCMC) algorithm. For the parsing model (Section 4.2), we derive 32 random sentences of length $n$ by randomly sampling words from a vocabulary of size 10000. As DPP utilizes a differentiable surrogate of the Eisner algorithm, it is compared to NES with the Eisner algorithm as the MAP solver. Similarly, SparseMAP is compared to NES with the CLE algorithm. Since our internal cluster consists of 4 GPUs, we utilize NES with $N = 4$ for a fair comparison with the gradient-based methods. Finally, we run the experiments over various random seeds and average the wall-clock time. Figure 2 depicts the results.

As can be seen, the run-times of DPP and SparseMAP heavily rely on the input dimension and grow at a much higher rate than the run-time of NES. NES also scales better than SST and REINFORCE on the NRI model. However, in this case, the gap is smaller as enlarging the input dimension of the latent structure also enlarges the model size which NES updates depend on. Overall, it can be seen that NES scales well with the latent space size in contrast to most of its competitors.

**Neural network size.** Next, we conduct a study that examines how the enlargement of a neural network affects the number of NES samples needed to optimize it. We begin by optimizing a VAE of 25K parameters, then we enlarge its parameter size by a factor of 2 and optimize the resulting model. We repeat this process several times up to a model of 800K parameters. We utilize SST with a fixed temperature of 1 as a baseline. For each model size, we examine how many NES samples are needed to achieve test ELBO as lower as the one achieved by the SST. Results are depicted in Table 3. A detailed description of the experimental setup can be found on Section B in the Appendix.

We observe that enlarging the neural network by a factor of 2 does not necessarily mean that we should enlarge $N$ in the same manner. To be precise, in all our experiments, we do not need to enlarge $N$ with more than 50% samples when optimizing the two times larger network. These observations are positive and suggest that NES can scale well with the network size.

## 5 RELATED WORK

Jang et al. [19], Maddison et al. [34] proposed the GSM estimator that replaces the non-differentiable $\arg\max$ operation with a differentiable *softmax* operation. However, structured latent spaces can be exponentially large and the softmax opertation becomes computationally intractable. Other works proposed tailor-maid solutions for specific structures. For instance, Corro & Titov [9] focused on latent projective dependency trees and propagated gradients through a differentiable surrogate of Eisner algorithm [11]. Mena et al. [36] extended the Gumbel-Softmax estimator [19] and proposed the Gumbel-Sinkhorn method for learning latent permutations. Paulus et al. [48] took these ideas one step further and proposed a unified framework for designing structured relaxations of combinatorial

Table 3: ELBO as a function of the neural network size and $N$. The values in the "Growth in $N$" column express the percent of additional NES samples (with respect to the preceding step) needed to achieve the baseline performance.

| # Parameters | SST | NES | $N$ | Growth in $N$ |
|---|---|---|---|---|
| 25K | -240.48 | -239.40 | 60 | - |
| 50K | -239.49 | -231.02 | 60 | + 0.00% |
| 100K | -233.13 | -232.40 | 90 | + 50.00% |
| 200K | -233.92 | -233.99 | 100 | + 11.11% |
| 400K | -239.02 | -234.29 | 100 | + 0.00% |
| 800K | -241.03 | -234.86 | 100 | + 0.00% |

distributions. Unlike these methods, our approach is generic and as such, it can be applied to general structures with no additional effort, since it obviates the need for a differentiable surrogate of the linear maximization oracle. Mensch & Blondel [37] proposed a framework for turning dynamic programming algorithms differentiable.

Others have taken a more generic approach. For example, SparseMAP [43, 42, 8] is a framework for training deep networks with sparse structured hidden layers, solved by sequential calls to a MAP oracle. In a similar sense, Itkina et al. [18] suggest using evidential theory to perform post hoc latent space sparsification and thus reducing the discrete latent sample space at test time. Chen et al. [5] generalized this method to a sparse normalization function which can be applied during both training and test time. Moreover, a recent line of works propagates gradients through the non-differentiable $\arg\max$ operation, Lorberbom et al. [33] use the difference of two maximization operations, and the method of Berthet et al. [1] is based on integration by parts. Contrarily, our approach does not require constructing sophisticated solutions to propagate gradients through discrete operations, which makes it both simple and flexible.

The Vector Quantized Variational Auto-Encoder (VQ-VAE) [56, 49] introduces an alternative approach to learning discrete latent representation. However, VQ-VAE differs from our discrete structured VAEs in an important aspect. In our setting, we know the structure of the latent space, e.g., the space of all possible spanning trees in a given graph. Hence we do not perform unsupervised vector quantization as in VQ-VAE but rather use a predetermined quantization over the set of possible structures. In this work, we rather focus on an alternative optimization method for learning discrete latent structures.

Recently, black-box optimization methods have been applied to neural networks [32, 52, 31, 61, 30, 53, 38]. Salimans et al. [52] showed that NES is a competitive alternative to popular RL techniques. Moreover, they utilized the fact that NES is highly parallelizable and proposed a generic distributed version of NES that scales well with the number of CPUs. Lenc et al. [31] proposed a hybrid method that alternates between NES and SGD for training large sparse models. Finally, Zhang et al. [61], Lehman et al. [30] compare the relation between the SGD gradients and NES updates. To our knowledge, we are the first to apply NES to structured VAEs.

## 6 CONCLUSION

We suggested using NES, a class of gradient-free black-box algorithms, as an alternative for learning discrete structured VAEs. We have demonstrated empirically that NES performs substantially better than various REINFORCE instances and even better than SST on the structure recovery task while being simpler and more robust. Moreover, NES achieves better or comparable performance to DPP and sparseMAP when considering dependency tree latent structure. However, as opposed to the aforementioned methods, NES does not require complex solutions for propagating gradients through the discrete structures, which makes it more generic, flexible, and simple to implement. Additionally, we showed that NES scales well with the latent space dimension and neural network size. To establish the theoretical soundness of our approach, we proved that NES converges for non-Lipschitz functions such as the objective function of a discrete VAE.

In this study, we have limited the expressive power of the NES method by fixing the covariance matrix of the Gaussian search distribution. For future work, we would like to explore the effect of jointly optimizing the covariance and mean of the distribution of parameters.

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

# A  PROOFS OF NES CONVERGENCE FOR NON-LIPSCHITZ FUNCTIONS

## A.1  PROOF OF LEMMA 1

*Proof.* The proof follows two main steps: (i) the spectral norm of $\nabla^2 g$ is at most $M/\sigma^2$ and (ii) $\|\nabla g(\mu_1)\|^2 \leq dM^2/\sigma^2$.

Since the Hessian matrix is symmetric, we can use Rayleigh quotient and obtain:

$$\nabla g(\mu_1)^\top \left(\nabla^2 g(\mu_2)\right)\nabla g(\mu_1) \leq |\lambda_{max}| \cdot \|\nabla g(\mu_1)\|^2, \tag{12}$$

where $\lambda_{max}$ is the largest eigenvalue of $\nabla^2 g(\mu_2)$. Since $\nabla^2 g(\mu_2)$ is symmetric it can also be shown that:

$$|\lambda_{max}| = \|\nabla^2 g(\mu_2)\| = \max_{\|s\|=1} |s^\top \nabla^2 g(\mu_2)s|. \tag{13}$$

Applying the log derivative trick on Eq. 6, we obtain

$$\nabla^2 g(\mu_2) = \int_{\mathbb{R}^d} \frac{1}{(2\pi)^{\frac{d}{2}}} e^{-\frac{\|w\|^2}{2}} \left(\frac{1}{\sigma^2} w w^\top\right) k(\mu_2 + \sigma w)dw. \tag{14}$$

Therefore:

$$
\begin{aligned}
\max_{\|s\|=1} |s^\top \nabla^2 g(\mu_2) s| &= \max_{\|s\|=1} \Big| \int_{\mathbb{R}^d} \frac{1}{(2\pi)^{\frac{d}{2}}} e^{-\frac{\|w\|^2}{2}} \Big( \frac{1}{\sigma^2} s^\top w w^\top s \Big) k(\mu_2 + \sigma w) dw \Big| \\
&\le \frac{M}{\sigma^2} \max_{\|s\|=1} \Big| \int_{\mathbb{R}^d} \frac{1}{(2\pi)^{\frac{d}{2}}} e^{-\frac{\|w\|^2}{2}} (s^\top w)^2 dw \Big| \\
&= \frac{M}{\sigma^2} \max_{\|s\|=1} \|s\|^2 = \frac{M}{\sigma^2}.
\end{aligned}
\tag{15}
$$

Since $k(\cdot)$ is bounded by $M$ and $w$ are i.i.d. normal Gaussian random variables, therefore for any $s$ the random $s^\top w$ is a Gaussian with zero mean and variance $\|s\|^2$. By combining Eq. 13 with Ineq. 15 we get:

$$
\|\nabla^2 g(\mu_2)\| \le \frac{M}{\sigma^2},
\tag{16}
$$

which concludes step (i). Next, we bound the squared norm of the gradient:

$$
\begin{aligned}
\|\nabla g(\mu_1)\|^2 &= \Big\| \int_{\mathbb{R}^d} \frac{1}{(2\pi)^{\frac{d}{2}}} e^{-\frac{\|w\|^2}{2}} \Big( \frac{w}{\sigma} \Big) k(\mu_1 + \sigma w) dw \Big\|^2 \\
&= \frac{1}{\sigma^2} \Big\| \int_{\mathbb{R}^d} \frac{1}{(2\pi)^{\frac{d}{2}}} e^{-\frac{\|w\|^2}{2}} w k(\mu_1 + \sigma w) dw \Big\|^2 \\
&\le \frac{1}{\sigma^2} \int_{\mathbb{R}^d} \Big\| \frac{1}{(2\pi)^{\frac{d}{2}}} e^{-\frac{\|w\|^2}{2}} w k(\mu_1 + \sigma w) \Big\|^2 dw \\
&\le \frac{M^2}{\sigma^2} \int_{\mathbb{R}^d} \frac{1}{(2\pi)^{\frac{d}{2}}} e^{-\frac{\|w\|^2}{2}} \|w\|^2 dw = \frac{dM^2}{\sigma^2},
\end{aligned}
\tag{17}
$$

where the first inequality is obtained using the Cauchy-Schwarz inequality and the second inequality by bounding $k(\cdot)$ with $M$. Thus, overall we have showed that:

$$
\begin{aligned}
\nabla g(\mu_1)^\top \nabla^2 g(\mu_2) \nabla g(\mu_1) &\le \frac{M}{\sigma^2} \cdot \|\nabla g(\mu_1)\|^2 \\
&\le \frac{dM^3}{\sigma^4}.
\end{aligned}
\tag{18}
$$

$\square$

## A.2 Proof of Theorem 1

*Proof.* Rearranging Eq. 9, we obtain:

$$
\eta \|\nabla g(\mu^t)\|^2 \le g(\mu^{(t)}) - g(\mu^{(t+1)}) + \eta^2 \frac{dM^3}{2\sigma^4}.
\tag{23}
$$

Summing over all algorithm steps $t = 1, \dots, T$, we have:

$$
\eta \sum_{t=1}^{T} \|\nabla g(\mu^t)\|^2 \le \sum_{t=1}^{T} [g(\mu^{(t)}) - g(\mu^{(t+1)})] + \eta^2 \frac{dM^3 T}{2\sigma^4}.
\tag{24}
$$

Opening the telescopic sum:

$$
\eta \sum_{t=1}^{T} \|\nabla g(\mu^t)\|^2 \le g(\mu^{(1)}) - g(\mu^{(T+1)}) + \eta^2 \frac{dM^3 T}{2\sigma^4}.
\tag{25}
$$

Since $k(\cdot)$ is non-negative and bounded, the difference between $g(\mu^{(1)})$ and $g(\mu^{(T+1)})$ is bounded from above by $M$:

$$\eta \sum_{t=1}^{T} \|\nabla g(\mu^t)\|^2 \leq M + \eta^2 \frac{dM^3 T}{2\sigma^4}. \tag{26}$$

We multiply both sides of the inequality by $\frac{1}{\eta T}$:

$$\frac{1}{T} \sum_{t=1}^{T} \|\nabla g(\mu^t)\|^2 \leq \frac{1}{\eta T} \left[ M + \eta^2 \frac{dM^3 T}{2\sigma^4} \right] \leq \frac{M}{\eta T} + \eta \frac{dM^3}{2\sigma^4}. \tag{27}$$

Next, we minimize the right-hand size of the inequality in $\eta$:

$$\eta^* = \sqrt{\frac{2\sigma^4}{T d M^2}}, \tag{28}$$

and plug it back to Ineq. 27:

$$\frac{1}{T} \sum_{t=1}^{T} \|\nabla g(\mu^t)\|^2 \leq \sqrt{\frac{2dM^4}{T\sigma^4}}, \tag{29}$$

Then, for arbitrarily small $\delta > 0$ such that:

$$\frac{1}{T} \sum_{t=1}^{T} \|\nabla g(\mu^t)\|^2 \leq \delta \leq \sqrt{\frac{2dM^4}{T\sigma^4}}, \tag{30}$$

there exists $t$ for which:

$$\|\nabla g(\mu^t)\|^2 \leq \delta, \tag{31}$$

after at most

$$T \leq \frac{2dM^4}{\delta^2 \sigma^4}, \tag{32}$$

steps. $\qquad\qquad\qquad\square$

## B  ADDITIONAL RESULTS

In the following experiments, we define the encoder as $input \Rightarrow MLP(\alpha) \Rightarrow ReLU \Rightarrow MLP(10) \Rightarrow argmax$, and the decoder as $MLP(\alpha) \Rightarrow ReLU \Rightarrow MLP(\beta) \Rightarrow output$, where $\beta$ is the input dimension. Unless otherwise stated, $\alpha = 300$.

### B.1  NEURAL NETWORK SIZE

The experiments were conducted on the FashionMNIST dataset [59] with fixed binarization [51]. We tune $\alpha$ for enlarging the VAE. Particularly, $\alpha$ is picked from the set $[16, 32, 64, 128, 256, 512]$. All models were trained using the ADAM optimizer [21] over 300 epochs with a constant learning rate of $10^{-3}$ and a batch size of 128.

### B.2  RELATION BETWEEN $g(\cdot)$ AND $k(\cdot)$

In section 3, we prove that under the conditions of Lemma 1, NES converges to a stationary point of $g(\mu)$ for non-Lipschitz functions. To empirically explore the relation between $g(\mu)$ and $k(\mu)$, we conduct a set of experiments in which we demonstrate that a low value of $g(\mu)$ is correlated with a low value of the objective function $k(\mu)$ by estimating the average absolute distance between them.

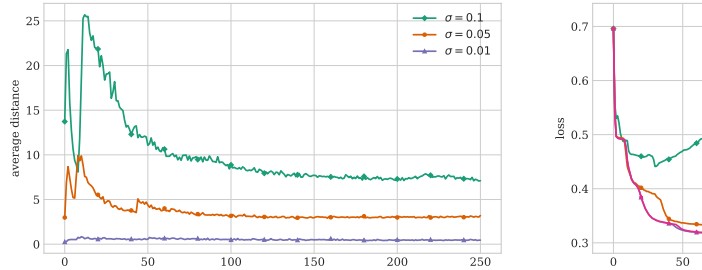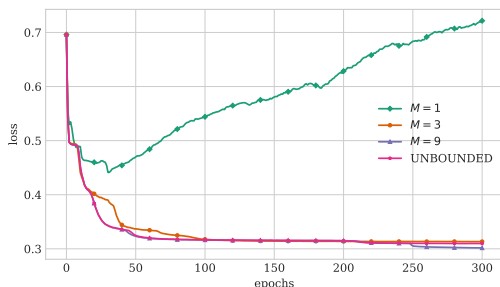

Figure 3: **Left:** The average absolute distance between the negative ELBO $k(\mu)$ and its Gaussian approximation $g(\mu)$ as a function of the training epoch. The smaller $\sigma$ is, the further the proximity between the two functions. **Right:** Negative ELBO as a function of the training epoch. Performance improves as the ELBO upper bound $M$ increases. Result suggests that bounding a discrete VAE loss with a large enough $M$ guarantees the convergence of NES within a finite number of iterations on the one hand, and on the other hand, does not impair performance.

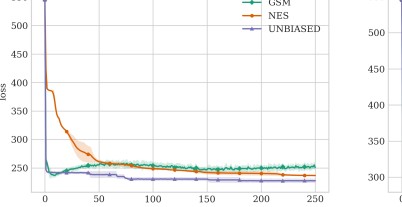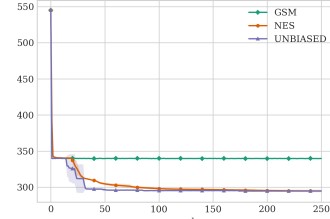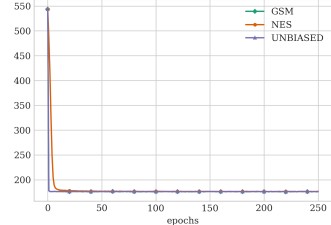

Figure 4: Negative ELBO as a function of the training epoch. Experiments are conducted on the binarized FashionMNIST (**Left**), KMNIST (**Center**), and Omniglot (**Right**) datasets. NES does not rely on computing gradients and yet achieves comparable performance with the unbiased and GSM methods. On the KMNIST benchmark NES even performs substantially better than GSM.

First, we estimate the Gaussian approximation $g(\mu)$ for each sample in the test set by perturbing the current model parameters 1000 times, computing the ELBO for each perturbed parameter vector and average. Then, we calculate the absolute difference between the ELBO, serving as the objective function, and the estimated Gaussian approximation and average over the tested samples. We experiment with three different NES configurations: $N = 300$ and $\sigma \in \{0.01, 0.5, 0.1\}$ on the FashionMNIST dataset [59]. The results presented on the left image in Figure 3 indicate that the smaller $\sigma$ is, the further the proximity between $k(\mu)$ and $g(\mu)$. It can also be seen that the average distance converges and stabilizes as the learning progresses towards saturation.

## B.3 BOUNDNESS ASSUMPTION

In the general case, the objective function of discrete VAEs is not bounded from above in contrast to Theorem 1 assumption. However, it can be upper bounded by bounding each log probability component with a constant. For ease of explanation, we scale the ELBO by dividing it with the VAE output dimension. Then, we upper bound it with $M = \{1, 3, 9\}$ during training and compare the test ELBO with that of a model trained with an unbounded ELBO, denoted by *UNBOUNDED*. We train the three models with $N = 300$ and $\sigma = 0.1$ on the FashionMNIST dataset. Results are depicted on the right image in Figure 3. It can be seen that bounding the loss has a minimal effect on model performance when $M$ is big enough. Increasing $M$ improves the performance, while using a relatively small $M$ value may cause the model to diverge. Surprisingly, bounding the loss with $M = 9$ leads to a slightly lower loss compared to the *UNBOUNDED* baseline. We hypothesize this is due to a regularization effect.

### B.4 UNSTRUCTURED TASKS.

Unlike structured VAEs, where the latent spaces are often exponentially large, here we explore a latent space that consists of only 10 different assignments. Therefore, an unbiased gradient of the objective with respect to the VAE parameters can be analytically computed by enumerating over all possible latent assignments (Eq. 1). We denote this method as *UNBIASED* and consider the loss of a model trained with this method as a lower bound for the loss of the same model trained with NES. Due to the relatively small latent space, we can also compare with a Gumbel-Softmax (GSM) biased estimator. For NES, the VAE is trained with $\sigma = 0.1$ and $N = 300$. For GSM, we use the annealing schedule of Jang et al. [19].

Experiments are conducted on the FashionMNIST [59], KMNIST [7], and Omniglot [29] datasets with fixed binarization [51]. All models are trained using the ADAM optimizer [21] with a constant learning rate of $10^{-3}$ and a mini-batch size of 128. Figure 4 depicts the negative ELBO of NES and its competitors.

Surprisingly, NES achieves competitive results compared to the *UNBIASED* method on all of the three benchmarks. On KMNIST, NES significantly outperforms GSM, and on the FashionMNIST and Omniglot benchmarks, it achieves comparable results. This is despite the fact that NES optimizes the VAE parameters by only evaluating the model at certain points in parameter space.

## C DERIVING EQUATION 2

Let $\gamma$ be a random function that associates an independent random variable $\gamma(z)$ for each input $z \in \mathcal{Z}$. When the random variables follow the Gumbel distribution law with mean $h_\phi(x, z)$, which we denote by $\mathcal{G}(h_\phi(x, z))$ and whose probability density function is $g_z(\gamma) = e^{-(\gamma(z)+c-h_\phi(x,z)+e^{-(\gamma(z)+c-h_\phi(x,z))})}$ for the Euler constant $c \approx 0.57$. Then for $g(t) = \prod_{z=1}^{k} g_z(t)$ we obtain the following identity:

$$e^{h_\phi(x,z)} = \mathbb{P}_{\gamma \sim g}[z^* = z], \text{ where } z^* \triangleq \arg\max_{\hat{z} \in \mathcal{Z}}\{\gamma(\hat{z})\}. \tag{33}$$

*Proof.* Let $G_z(t) = e^{-e^{-(t+c-h_\phi(x,z))}}$ be the Gumbel cumulative distribution function. Then

$$\mathbb{P}_{\gamma \sim g}[z^* = z] = \mathbb{P}_{\gamma \sim g}[z = \arg\max_{\hat{z}=1,\ldots,k}\{\gamma(\hat{z})\}]$$
$$= \int g_z(t) \prod_{\hat{z} \neq z} G_{\hat{z}}(t))dt. \tag{34}$$

Since $g_z(t) = e^{-(t+c-h_\phi(x,z))}G_z(t)$ it holds that

$$\int g_z(t) \prod_{\hat{z} \neq z} G_{\hat{z}}(t)dt = \int e^{-(t-h_\phi(x,z)+c)}G_z(t) \prod_{\hat{z} \neq z} G_{\hat{z}}(t)dt$$
$$= \frac{e^{h_\phi(x,z)}}{Z}, \tag{35}$$

where $\frac{1}{Z} = \int e^{-(t+c)} \prod_{\hat{z}=1}^{k} G_{\hat{z}}(t)dt$ is independent of $z$. Since $\mathbb{P}_{\gamma \sim g}[z = z^*]$ is a distribution then $Z$ must equal to $\sum_{\hat{z}=1}^{k} e^{h_\phi(x,\hat{z})}$. $\qquad\square$

Next, we use the Gumbel-Max trick to rewrite the expected log-likelihood in the ELBO in the following form:

$$\mathbb{E}_{z \sim q_\phi} \log p_\theta(x|z) = \sum_{z \in \mathcal{Z}} \mathbb{P}_{\gamma \sim g}[z^* = z]f_\phi(x, z) = \mathbb{E}_{\gamma \sim g}[f_\theta(x, z^*)]. \tag{36}$$

The equality results from the identity $\mathbb{P}_{\gamma \sim g}[z^* = z] = \mathbb{E}_{\gamma \sim g}[1_{z^*=z}]$, the linearity of expectation $\sum_{z \in \mathcal{Z}} \mathbb{E}_{\gamma \sim g}[1_{z^*=z}] f_\phi(x, z) = \mathbb{E}_{\gamma \sim g}[\sum_{z \in \mathcal{Z}} 1_{z^*=z} f_\phi(x, z^*)]$ and the fact that $\sum_{z \in \mathcal{Z}} 1_{z^*=z} = 1$.

When $z = (z_1, ..., z_{|E|})$ is a spanning tree, or more generally, belongs to the a structured space, one cannot assign an i.i.d. random variable to each $z \in \mathcal{Z}$. Instead we relate a random variable $\gamma = (\gamma_1, ..., \gamma_{|E|})$ and set $\gamma(z) = z^\top \gamma$.

## D    THE FULL OBJECTIVE FUNCTION

In Eq. 7 we didn't include the KL-divergence term for the sake of simplicity. In practice, the NES algorithm optimizes both terms. Thus, for completeness we provide the full NES objective. For the avoidance of doubt, in our experiments, we optimized both terms.

Assuming that $p_\theta(\cdot)$ is the uniform distribution over the space of structures, the Gumbel-Max reparameterization trick let us derive the following approximation:

$$KL(q_\phi(\cdot|x)||p_\theta(\cdot)) = \mathbb{E}_{z \sim q_\phi(\cdot|x)}\Big[\log \frac{q_\phi(z|x)}{p_\theta(z)}\Big] \approx \mathbb{E}_{\gamma \sim \mathcal{G}(h_\phi(x))}[z^{*\top} h_\phi(x) - \log \frac{1}{|\mathcal{Z}|}], \quad (37)$$

where $z^* = \arg\max_{z \in \mathcal{Z}}\{z^\top \gamma\}$. The resulting NES objective is:

$$\mathbb{E}_{w \sim N(\mu, \sigma^2 I)} \mathbb{E}_{\gamma \sim \mathcal{G}(h_{w_2}(x))}\Big[-f_{w_1}(x, z^*) + z^{*\top} h_{w_2}(x) - \log \frac{1}{|\mathcal{Z}|}\Big]. \quad (38)$$

And its gradient takes the form of:

$$\mathbb{E}_{w \sim N(0, I)} \mathbb{E}_{\gamma \sim \mathcal{G}(h_{\mu_2 + \sigma w_2}(x))}\Big[\frac{w}{\sigma}\Big(-f_{\mu_1 + \sigma w_1}(x, z^*) + z^{*\top} h_{\mu_2 + \sigma w_2}(x) - \log \frac{1}{|\mathcal{Z}|}\Big)\Big], \quad (39)$$

where $w = [w_1; w_2]$ is the concatenation of the two vectors $w_1, w_2$.

