# OpenReview forum: "Learning Discrete Structured Variational Auto-Encoder using Natural Evolution Strategies"
_ICLR.cc/2022/Conference — ICLR 2022 Poster_

### Official Review · Reviewer_qpkv · 2021-11-01

**Correctness:** 4
**Technical Novelty And Significance:** 4
**Empirical Novelty And Significance:** 4
**Recommendation:** 8
**Confidence:** 4

**Main Review:**

This is an elegant paper.  Although, NES algorithms have been used elsewhere, this is the first application to VAE training.  By demonstrating its feasibility and effectiveness the authors highlight the usefulness of the approach.  The discussion and analysis is informative and of a high scientific standard.  The use is the approach in this contexts requires the use of a number of sampling tricks which makes the approach quite subtle.

One could argue that this is just an application of NES to yet another application, but I think this would be unfair.  The REINFORCE algorithm has become a important, goto, algorithm in deep-learning.  Showing that NES provides a simple and efficient algorithm for solving non-differentiable problems is an important contribution worth making.

The paper is well written, accurate and to the point.

**Summary Of The Paper:**

The paper proposes and analyses the use of Natural Evolutionary Strategies (NES) to train a VAE with a discrete latent space representation (Spanning Trees).  NES provides a smoothed approximation of the discrete function for which the gradient can be easily computed.  This approach can be seen as an example of the REINFORCE algorithm, although it clearly shows superior performance to traditional REINFORCE implementations.  The authors show that not only is it competitive and often marginally better than state-of-the-art techniques for this problem, it is also much more straightforward and easier to generalise.

**Summary Of The Review:**

This is in my view a good paper that deserves a place in ICLR.

---

> ### Author Response · Authors · 2021-11-15
> **Response to Reviewer qpkv**
>
> We thank reviewer qpkv for the positive feedback. \
> We are glad the reviewer thinks that our paper is elegant and of a high scientific standard, as much thought went into it. \
> Like the reviewer, we believe that showing that NES can solve a challenging non-differentiable problem (like optimizing discrete structured VAE) in an effective yet simple manner is an important contribution to the community.

---

### Official Review · Reviewer_ANuT · 2021-11-01

**Correctness:** 4
**Technical Novelty And Significance:** 3
**Empirical Novelty And Significance:** 2
**Recommendation:** 8
**Confidence:** 4

**Main Review:**

The strengths of the paper:
- proposing another approach to optimization a neural network than through propagating gradients,
- proving that NES converges for non-Lipschitz functions such as the objective function of a discrete VAE,
- experimentally showing that gradient-free methods have similarly effective as perturb-and-parse gradient-based methods

The weaknesses of the paper:
- optimizing only the mean of the distribution of parameters, a covariance matrix of the distribution of parameters is imposed by the user. The classical model VAE optimizes both these parameters.
- too poor experiments, the authors showed experiments that presented effective optimization of the ELBO values and how the latent space size affects the method’s run-time. It could be possible to show how the model reconstructs the images or generates them (this can be troublesome due to the lack of optimization of the VAE model variance). The results of such experiments can be compared to the results of the VQ-VAE method (https://arxiv.org/pdf/1711.00937.pdf or VQ-VAE 2 - https://arxiv.org/pdf/1906.00446.pdf).

**Summary Of The Paper:**

In this paper, the authors proposed to use the Natural Evolution Strategy (NES) algorithm for learning discrete structured VAEs. This algorithm estimate gradients with forwarding pass evaluations only and do not require propagating gradients through their discrete structures. Authors showed empirically that optimizing discrete structured VAEs using NES is as effective as gradient-based approximations.

**Summary Of The Review:**

The paper is well written, the authors clearly describe the problem and the proposed solution that is computationally appealing because does not require propagating gradients. It is theoretical work, the experiments only show the effectiveness of the ELBO function optimization. The authors consider only optimization of the mean distribution of parameters and do not show other experiments that confirm the effectiveness of such optimization in generative models (see the section of the weaknesses of the paper). Hense I rate the work good but not sufficient for the ICLR conference.

---

> ### Author Response · Authors · 2021-11-15
> **Response to reviewer ANuT**
>
> We thank reviewer ANuT for the comments and feedback. We are glad the reviewer thinks the paper is well written and that the problem and proposed solution are clearly described.
>
> In the following, we address the reviewer concerns:
>
> **Learning the covariance matrix**
>
> We thank the reviewer for emphasizing learning of the covariance matrix. We left such learning out to follow standard discrete VAEs baselines (e.g.,  [1, 2, 3, 4], Gumbel-Softmax, Gumbel-Sinkhorn, SparseMAP, Stochastic Softmax Tricks, Perturb and Parse) that learn only the mean parameters. We feel that comparing to baselines that do not naturally embed variance is unfavorable to the baselines.
>
> Nevertheless, we conducted a set of experiments that show that learning the variance further improves NES performance. We used the VAE architecture described in Section B in the Appendix. The experiments were conducted on the binarized MNIST, FashionMNIST, and KMNIST datasets. The model was trained using the ADAM optimizer over $300$ epochs with $N=150$ and a constant learning rate of $10^{-3}$ for learning the mean and $5\cdot 10^{-5}$ for learning the covariance. We compared the performance of NES when learning both the mean and covariance to the performance of NES when learning only the mean. In the following table, we report the negative ELBO of the different experiments (lower is better):
>
> | Method \ Dataset        |  MNIST | FashionMNIST | KMNIST |
> |-------------------------|:------:|:------------:|:------:|
> | NES (mean)              | 167.12 |    225.36    | 293.35 |
> | NES (mean + covariance) | 166.23 |    224.09    | 291.63 |
>
> As can be seen, jointly optimizing the mean and covariance leads to a minor improvement. We leave further exploration for future work.
>
> **Reconstructing/Generating images**
>
> In our paper, we focused on VAEs whose latent space is discrete and structured. In our experimental validation we focused on the latent space of spanning trees and the challenge is to choose a spanning tree out of exponentially many spanning trees. We conducted a comprehensive comparison between NES and the leading methods for optimizing such systems and used relevant metrics such as edge precision, edge recall, unlabeled attachment score, run-time, and the ELBO to measure performance.
>
> In addition, we investigated some properties of NES in a controlled setting, i.e., the effect of latent space size on model runtime, the effect of model size on the number of samples needed for optimization, etc. We don’t claim that NES leads to better (or worse) image reconstruction compared to the baselines, as it’s not the essence of our work. The main objective of our experiments was to measure the ability of NES in learning latent structures compared to gradient-based methods. Quoting reviewer 12Bq: “The experiments are extensive and convincing, demonstrating advantages over a wide range of baselines.”
>
> Regarding the comparison to VQ-VAE, we agree that there is a similarity between the methods (both methods perform auto-encoding under the discrete variational setting), however, the goal is different. In our setting we **know** the structure of the latent space, e.g., the space of all possible spanning trees in a given graph, hence we do not perform unsupervised vector quantization as in VQ-VAE but rather use a predetermined quantization over the possible spanning trees. Comparing our method to VQ-VAE is unfavorable to VQ-VAE as its unsupervised vector quantization never results in valid spanning trees.
>
> [1] Jang, Eric, Shixiang Gu, and Ben Poole. "Categorical Reparameterization with Gumbel-Softmax", in ICLR, 2017. \
> [2] Mena, Gonzalo, et al. "Learning latent permutations with gumbel-sinkhorn networks", in ICLR, 2018. \
> [3] Niculae, Vlad, et al. "Sparsemap: Differentiable sparse structured inference", in ICML, 2018. \
> [4] Corro, Caio, and Ivan Titov. "Differentiable perturb-and-parse: Semi-supervised parsing with a structured variational autoencoder", in ICLR, 2019. \
> [5] Paulus, Max B., et al. "Gradient estimation with stochastic softmax tricks", in NeurIPS, 2020.

---

### Official Review · Reviewer_Fqt8 · 2021-11-02

**Correctness:** 3
**Technical Novelty And Significance:** 3
**Empirical Novelty And Significance:** 3
**Recommendation:** 6
**Confidence:** 3

**Main Review:**

Strengths.

1. The paper is well-organized.

2. In this work, a gradient-free black-box optimization algorithm is explored for discrete structured VAEs.
They also experimentally show gradient-free methods are as effective as sophisticated gradient-based methods.

3. They prove that the NES algorithm converges for non-Lipschitz functions. This is different from the contemporary trend that relies on Lipschitz functions.

Weaknesses
1. In VAES, there are two terms for ELBO: one is sampled log-likelihood term, the other is the KL-divergence term that measures the similarity of the auxiliary distribution q and the unknown distribution p.  A lot of people notice the importance of the second term. In this paper, the authors use gradient-free methods for the first term. However, the second term is missing in the algorithm. Or do I miss something?

2. The gradient-free black-box optimization algorithm is generally with high variance (for example, refinance force algorithm). Unfortunately, there are no clear discussions on this. And how is the effect of the proposed method? Mirrored sampling is used in the experimental setup. It is not clear whether this sampling method is used in other methods. This leads to a question: whether there is a fair comparison.


**Summary Of The Paper:**

In this paper, the authors use Natural Evolution Strategies (NES) to learn discrete structured VAE. They empirically show that optimizing discrete structured VAEs using NES is as effective as gradient-based approximations. And they also prove that NES converges for non-Lipschitz functions in discrete structured VAEs.

**Summary Of The Review:**

It would be great to provide more details on how to get Equation (2). Especially, why an argmax operation is added.

---

> ### Author Response · Authors · 2021-11-15
> **Response to Reviewer Fqt8**
>
> We thank reviewer Fqt8 for the questions and feedback. \
> In the following, we address the reviewer concerns:
>
> **Optimizing the KL-divergence term**
>
> The reviewer is correct. We didn’t include the KL-divergence term in the algorithm for the sake of simplicity. In practice, the NES algorithm optimizes both terms. For the avoidance of doubt, in our experiments, we optimized both.
>
> For completeness we provide the full objective function: \
> Assuming that $p_\theta (\cdot)$ is the uniform distribution over the space of structures, the Gumbel-Max reparameterization trick let us derive the following approximation
>
> $
> KL(q_\phi (\cdot|x) || p_\theta(\cdot)) = E_{z \sim q_\phi (\cdot|x)} \big [\log \frac{q_\phi (z|x)}{p_\theta (z)} \big ] \approx E_{\gamma \sim \mathcal{G}(h_\phi(x))} [z^{*^{\top}} h_{\phi}(x) - \log \frac{1}{|\mathcal{Z}|}],
> $
>
>
> where $z^* = \arg\max_{z \in \mathcal{Z}} \{z^{\top}\gamma\}$. The resulting NES objective is
>
> $E_{w \sim N(\mu, \sigma^2 I)} E_{\gamma \sim \mathcal{G}(h_{w_2}(x))} \big [ - f_{w_1}(x, z^*) + z^{*^{\top}} h_{w_2}(x) - \log \frac{1}{|\mathcal{Z}|} \big ]$.
>
> and its gradient takes the form of
>
> $E_{w \sim N(0, I)} E_{\gamma \sim \mathcal{G}(h_{\mu_2 + \sigma w_2}(x))} \Big [\frac{w}{\sigma} \big( -f_{\mu_1 + \sigma w_1}(x, z^*) + z^{*^{\top}} h_{\mu_2 +\sigma w_2}(x) - \log \frac{1}{|\mathcal{Z}|} \big ) \Big ]$,
>
> where $w = (w_1, w_2)$ is the concatenation of the two vectors $w_1, w_2$.
>
> We thank the reviewer for this important question that highlights the need to include these equations. We will include them in the supplementary material in the updated version, which we will upload in a few days.
>
> **The variance of NES**
>
> As stated by the reviewer, the score function estimator (REINFORCE) is known to suffer from high variance (as discussed in the background section). NES is an instance of REINFORCE that optimizes a smoothed version of the objective function by sampling from a distribution over parameter space (rather than the latent space). This distribution is commonly modeled as a multivariate Gaussian. A common practice for reducing the variance of NES is using the mirrored sampling technique where we use a single Gaussian noise vector to create two parameter sets, one by adding and the other by subtracting the noise vector. This technique is only possible for symmetric distributions (e.g., Gaussian) and unfortunately cannot be applied to  our baselines, as unlike NES, they are not based on learning a Gaussian distribution over the space of the parameters.
>
> Nevertheless, various general techniques were devised to reduce REINFORCE variance in our baselines. In the latent structure recovery experiments (Section 4.1), we compare NES performance against the performance of four instances of REINFORCE, where each utilizes a different variance reduction technique. The results as described in Table 1 show that the variance of NES is low and, in some cases, even negligible compared to the variance of its competitors.
>
> **How to derive Equation 2**
>
> We will provide the steps to derive Equation 2 in the supplementary material.

---

### Official Review · Reviewer_12Bq · 2021-11-03

**Correctness:** 3
**Technical Novelty And Significance:** 3
**Empirical Novelty And Significance:** 3
**Recommendation:** 8
**Confidence:** 3

**Main Review:**

**Strengths:**
* The paper addresses an open problem in the space of discrete VAEs.
* The paper is very well written, easy to follow, and compelling.
* The presentation of the background material logically leads the reader into the proposed methodology, and the math is included as a meaningful and coherent addition to the story.
* The experiments are extensive and convincing, demonstrating advantages over a wide range of baselines.

**Weaknesses and suggestions for improvement:**
* Some relevant works are missing from the literature review [1-6]. Contextualizing the proposed method in these works would make the overall paper stronger.
* Since Eq. (2) is critical to the proposed method and the presented theoretical results, it would be helpful to highlight its importance when it is first defined. I recommend you clearly state that Eq. (2) is the objective function you will consider for the remainder of the paper, and elaborate on why it is non-negative (i.e., is it because of a ReLU at the end of the $f_{\theta}$ network?). Additionally, please highlight in the proof of Lemma 1 where the non-negativity comes in.
* I would appreciate the inclusion of a discussion of the bias introduced by the NES algorithm in Eq. (4). Further discussion on why $N$ forward passes in NES are more desirable than a sampling-based technique with $N$ samples would also be helpful. The experiments and the parallelization arguments contribute to this discussion, but a bit more emphasis and detail in the methods section would aid the reader to better understand the advantages of the proposed method.
* I recommend the authors spend a bit more time explaining what $w_{1}$ and $w_{2}$ correspond to, i.e., the encoder and decoder parameters, respectively? At first read, I thought you were setting $N = 2$.
* The statement on pages 4 and 5 that for any $T$, there exists a $t \in \{1, ..., T\}$ for which the magnitude of the gradient is arbitrarily small needs to be corrected. Specifically, it should say something like for *T sufficiently large* the magnitude of the gradient is arbitrarily small.
* In the experiments, additional discussion would be helpful for why SST might be expected to do better than NES in Table 1, and the disadvantages of SparseMap compared to NES.
* Lastly, the paper needs to be proofread for typos, as I found a number of them while reviewing. Some examples include: 'over exponentially large latent space', missing period after Eq. (3), 'using NES algorithm' (missing 'the'), incorrect quotation marks for 'parameters' and "Growth in N", 'in contrast to contemporary trend' (missing 'the'), 'which must carefully designs', 'being simpler and robust' (missing 'more'), 'generic, flexible, and simpler to implement' (missing 'more'). The references need to be proofread for consistency, as well: some venues are missing, some venues are only presented as acronyms, while others are presented in full, incorrect capitalization ('Ai'), etc. If possible, avoid splitting equations across multiple lines in the text and capitalize 'Section' when referencing section numbers.

[1] A. van den Oord, O. Vinyals, and K. Kavukcuoglu, "Neural discrete representation learning," in NeurIPS, 2017.

[2] A. Razavi, A. van den Oord, and O. Vinyals, "Generating diverse high-fidelity images with VQ-VAE-2," in NeurIPS, 2019.

[3] G. Correia, V. Niculae, W. Aziz, and A. Martins, "Efficient marginalization of discrete and structured latent variables via sparsity," in NeurIPS, 2020.

[4] M. Itkina, B. Ivanovic, R. Senanayake, M. J. Kochenderfer, and M. Pavone, “Evidential sparsification of multimodal latent spaces in conditional variational autoencoders,”in NeurIPS, 2020.

[5] P. Chen, M. Itkina, R. Senanayake, and M. J. Kochenderfer, "Evidential softmax forsparse multimodal distributions in deep generative models," in NeurIPS, 2021.

**Summary Of The Paper:**

The proposed paper presents a method for optimizing discrete structured VAEs using Natural Evolution Strategies (NES). The authors present a theoretical conclusion regarding algorithm convergence and extensive empirical results demonstrating that NES performs comparably to gradient-based methods, while being more computationally efficient.

**Summary Of The Review:**

Overall, the authors present an interesting method for optimizing discrete VAEs with compelling theoretical and experimental results. The paper is well written and the story is coherent. My recommendation is to accept.

---

> ### Author Response · Authors · 2021-11-15
> **Response to Reviewer 12Bq**
>
> We thank reviewer 12Bq for the constructive feedback and the detailed suggestions. We will update the paper accordingly and upload a revised version within a few days. The updates will be color-marked.

---

### Author Response · Authors · 2021-11-22
**A revised version has been submitted**

We thank all reviewers for the valuable feedback we have received. The paper was revised accordingly, and the modifications were marked in blue. We have made the following changes:

•	Added the related works mentioned by reviewer 12Bq \
•	Emphasized the importance of Eq. (2) when it is first defined \
•	Explicitly described $f_{\theta}$ \
•	Described where we use the non-negativity of $k(\cdot)$ in the proof of Theorem 1 \
•	Discussed the bias introduced by the NES algorithm \
•	Discussed why $N$ forward passes in NES are more desirable than a sampling-based technique with $N$ samples \
•	Added an explanation about what $w_1$ and $w_2$ correspond to \
•	Corrected the theoretical statements on pages 4 and 5 \
•	Explained why SST might be expected to do better than NES \
•	Emphasized the disadvantage of SparseMAP compared to NES \
•	Fixed the typos \
•	Added a Section describing the full objective function (which includes the KL-divergence term) to the Appendix \
•	Added the derivation of Eq. (2) to the Appendix

If there are any questions left, please let us know before the discussion period ends.

---

### Decision · Program_Chairs · 2022-01-20

**Decision:**

Accept (Poster)

**Comment:**

The authors propose to use genetic algorithms to learn variational autoencoders (VAEs) with discrete latent spaces. Specifically they employ natural evolution strategies (NES) to avoid backpropagating gradients through discrete variables. Experiments show how the proposed approach is competitive with the current state-of-the-art to train discrete VAEs.

Some concerns arose from the review and discussion phases, these included confusion around the justification and derivation of NES for VAEs in the presentation and the limitation of the experiments. Authors were responsive and provided the reviewers the needed clarifications, an updated presentation in the revised paper and additional experimental results which ultimately were successful in raising the reviewers' scores towards full acceptance.